# Adaptive Stretch-Forming Process: A Computer Vision and Statistical Analysis Approach

**Cosmin Constantin Grigoras** [1], **Valentin Zichil** [1,*], **Bogdan Chirita** [2] and **Vlad Andrei Ciubotariu** [2]

1   Department of Engineering and Management, Mechatronics, "Vasile Alecsandri" University of Bacău, 157 Calea Mărăşeşti, 600115 Bacau, Romania; cosmin.grigoras@ub.ro
2   Department of Industrial Systems Engineering and Management, "Vasile Alecsandri" University of Bacău, 157 Calea Mărăşeşti, 600115 Bacau, Romania; chib@ub.ro (B.C.); vlad.ciubotariu@ub.ro (V.A.C.)
*   Correspondence: valentinz@ub.ro

**Abstract:** An industrial process is defined through its quality of parts and their production costs. Labour-intensive operations must be applied to produce high-quality components with inexpensive resources. Recent development in dedicated software allows the industrial sector to rely on more and more autonomous solutions to obtain an optimum ratio between part quality and cost. The stretch forming process is an operation that has a high degree of difficulty, due to the process parameters and the spring-back effect of materials. Our approach to solving several of the shortcomings of this process was to develop a self-adaptive algorithm with computer vision capabilities that adapts to the process in real-time. This experimental study highlights the results obtained using this method, as well as a comparison to a classical method for the stretch-forming process (SFP). The results have noted that the stretch-forming algorithm improves the process, while adapting its decisions with each step.

**Keywords:** computer vision; statistical analysis; adaptive stretch-forming; 3D measurement system

## 1. Introduction

In its simplest form, stretch-forming involves applying biaxial tension on a metal sheet [1], as indicated in Figure 1. The material must be stretched in the axial direction as a die pushes, in a perpendicular direction, drawing the metal sheet into the desired shape [2–4]. This process gradually produces severe plastic deformation (SPD) due to the increased stress; therefore, strain distribution must be considered [5]. As a result, the strain increases by a specific amount, depending on the mechanical properties of each material [6–12]. Another aspect is the deformation, since materials behave differently when elastic or plastic deformation occurs [2,13]. In the elastic domain, materials follow Hooke's law with their predictable behavior, with a constant slope between stress and strain (Young's modulus). In the plastic domain, the Theory of Elasticity indicates that more complex phenomena occur [14,15].

This complex process is used in industrial sectors, such as aviation, automation, rail transport, or architecture. Due to the increasing demands of vehicles, aircraft, or high-speed trains with low fuel consumption and electric capabilities, stretch-forming is used to manufacture large parts that require fewer assembly components, aiming to reduce the overall weight [16–20]. The architectural sector uses the SFP for complex shape panels that offer an organic shape to buildings' interior or exterior [21].

Numerous studies in this field have been conducted [22]. The implementation of this metal sheet forming process is often highlighted in scientific studies as finite element analysis [2,19,23]. Successful implementations of new concepts are assigned to processes such as multi-point die stretch-forming (MPD-SF) [24], in combination with single point incremental forming (SF-SPIF) [22] or electromagnetic incremental forming (EIF) [25]. This extensive research is directed on the uniform-contact state [20], the effects of friction

on the stretch-forming limit curves [1], industrial equipment improvements [7,26,27], or process design and effects [26,28] on the deformability. Experimental or numerical studies are mainly based on the analysis of metallic materials that are related to the automotive and aviation sectors; as a result, aluminum (AA6061-T4 [2], AA6082 [18], AA2024, AA1050, [29,30]), magnesium (LZ91, AZ31, ZE10, ME21 [31,32]), titanium (OT04, Ti-6Al-4V [31–33]) or steel (CR1 grade [34])-based alloys are intensively researched.

Furthermore, the complexity of the stretch-forming process implies computer-aided techniques for in-process control, predictions of deformability, spring-back effect, final shape, and dimensions of parts by analyzing the relationship of the process parameters. These complex analyses are assigned to custom software that implies using neural network (NN) k (NN) [35], deep neural networks (DNNs) [36], deep learning (DL) [37], support vector regression (SVR) [38], and computer vision-based algorithms (CV) [39].

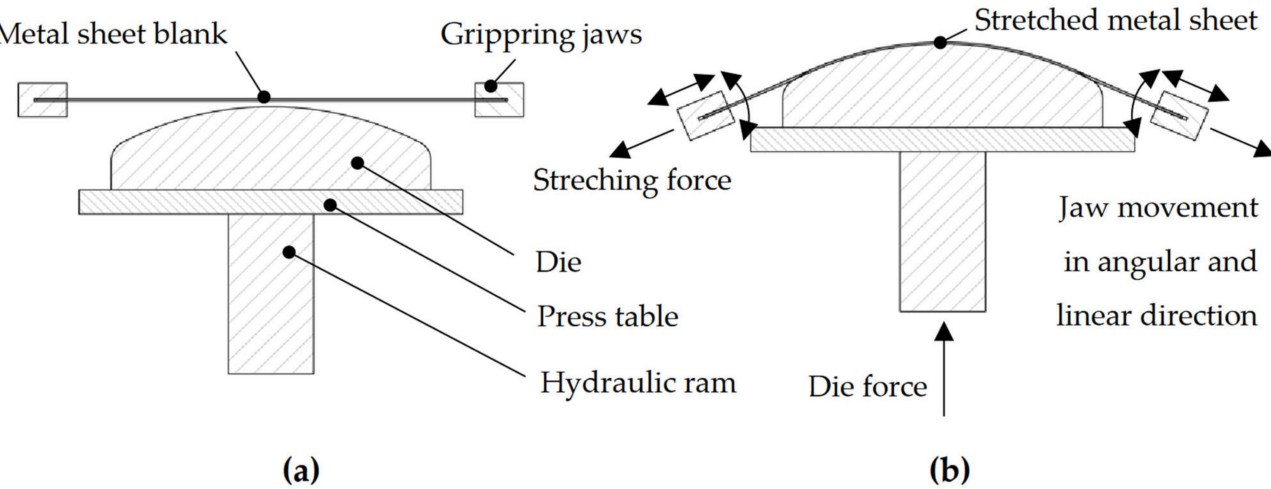

**Figure 1.** (**a**) Schematic representation of the transversal stretch-forming process indicating the main components of the process; (**b**) Stretch-forming process working principle highlighting the deformed shape of the metal sheet, die movement (die force) in the vertical direction, stretching force, and the linear and angular jaw movement.

## 2. ASFP Algorithm and Industrial Setup

The Adaptive Stretch-Forming Process (ASFP) is a self-adaptive video and statistical analysis algorithm. It is, at its core, a software tool that calculates the strain, deciding and evaluating its choice so that it can control a hydraulic pump in real-time, ensuring axial tension, as well as a hydraulic press, ensuring the die movement, thus obtaining control of the biaxial nature of the process, as described in Figure 2. It does this by constantly reading the position of two round markers with known dimensions, placed on specific areas on the material blank. The software, written in Python, uses libraries such as OpenComputerVision (OpenCV) [40] for computer vision, numerical Python (NumPy) [41] for numerical operations, statsmodels (statsmodels.api and statsmodels.formula.api) [42] for statistical analysis, SerialPy [43] for serial communication with the microcontrollers, and MathPlotlib [44] for graph rendering. The ASFP algorithm can run in both programmed and software-controlled (autonomous) modes. The first case only measures the strain, while the autonomous version decides on how to control the industrial equipment according to the limit values indicated by the user. This can be a strain limit or specific stress, if the correlation of the stress on strain curve values is taken into consideration or both stress and strain values; the goal is to avoid material failure while reaching a high degree of deformation.

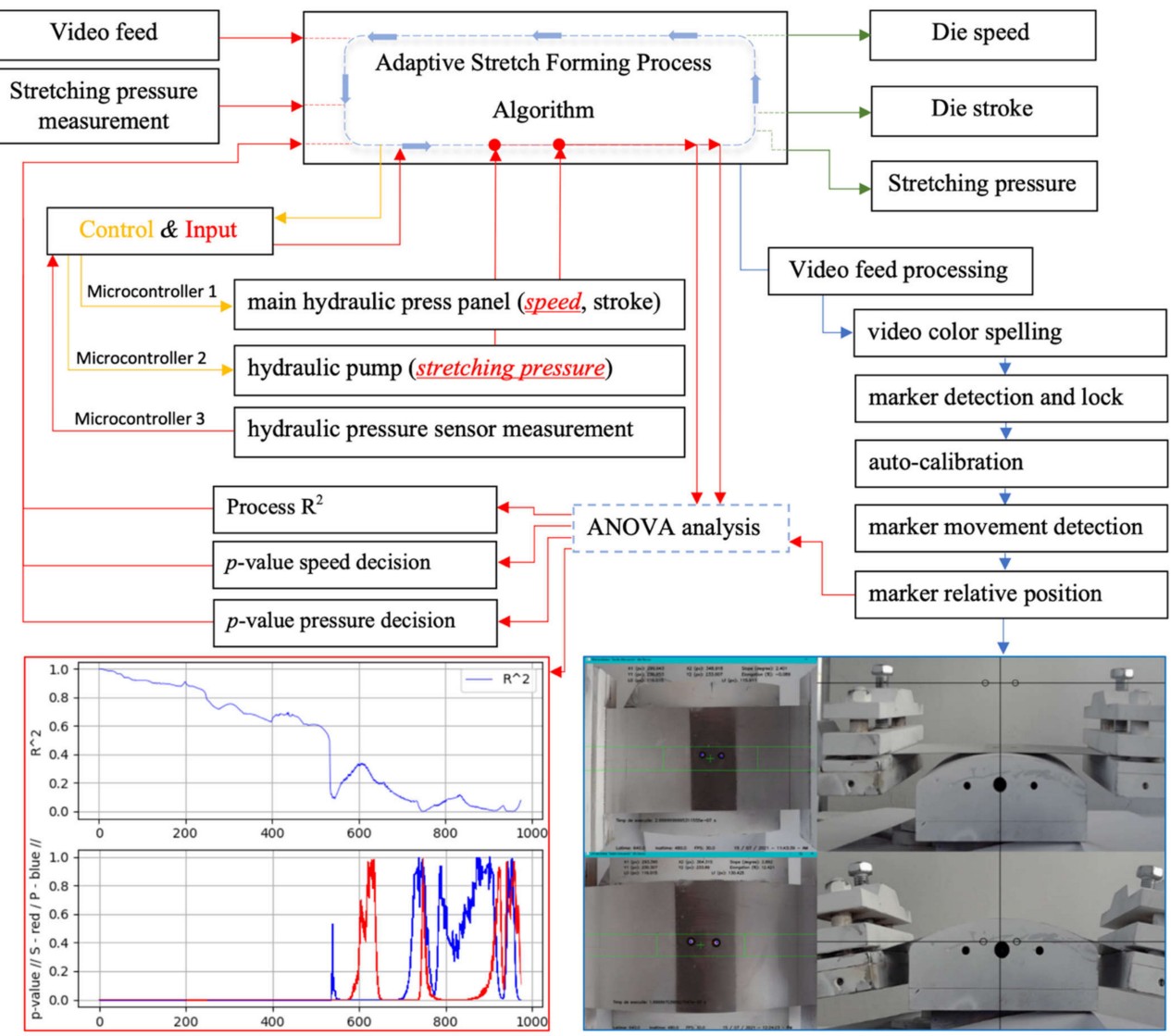

**Figure 2.** Schematic representation of the Adaptive Stretch Forming Process (ASFP) algorithm logic, indicating the input data sources, output controlled process parameters, decision-making methodology, processed and side view video feed along with graphs for the $R^2$ process, decision *p*-value for die speed, and stretching pressure.

The algorithm makes a pre-analysis of the central section of the video feed, expecting to find the two round markers; if this step is successful, two distinct decisions take place: the industrial equipment is turned on, usually assigning an initial die speed between 0.09 and 1.08 mm/s, gradually increasing the stretching pressure from 0 bar, and processing the video feed frame by frame, as described below in Table 1. Once the process starts, some decisions (die speed and stretching pressure) are given to the industrial equipment and stored in csv files along with strain values; these data are then statistically analyzed by using the ANOVA method so that the algorithm adapts and adopts the best decisions constantly comparing the actual coefficient of determination (R-squared/$R^2$) value to the previous one. The *p*-values offer another method of comparison for speed and pressure. If the actual decision has the same or higher significance with the previous one, the program makes no changes. Otherwise, if the results are lower than expected, compared to the previous ones, the die speed and the stretching pressure are being constantly adjusted and the decisions compared. The goal is to improve as much as possible the *p*-value for the speed and pressure decisions and, subsequently, the R-squared process value, if this can be done.

**Table 1.** The ASFP algorithm for the video-feed processing steps.

| Video Processing Description | |
| --- | --- |
| video colour spelling | image colour spelling [45] from red-blue-green to black and white for edge boundary detection; |
| marker detection and lock | the material blanks are sprayed with an anti-reflex coating; if any residual points (light reflections, marks on the part, round corners of the die) still appear, they are cancelled by the software; at this step, by using the Hough circle transformation method [46], the two markers are locked into position and only they are analyzed; |
| auto-calibration | calibrated marks with a fixed diameter (white on black round markers with a radius of 1.5 and 3.5 mm) are used; the auto-calibration algorithm sets the necessary numerical value of the calibration factor; |
| marker movement detection | the Lucas-Kanade track and trace optical flow method [47] is used to analyze the position of each marker with each frame; |
| marker relative position | the position is read as the distance between pixels by using the lower-left corner of the video feed as the origin; the strain is calculated as percentage displacement from the initial to the actual position %. |

The experimental setup, highlighted in Figure 3a, consists of three components: industrial equipment, input data systems, and control systems. The industrial equipment consists of a Hydramold hydraulic press (Hydramold, Iași, Romania) and an Ecoroll HGP 3.0 hydraulic pump (Ecoroll AG, Celle, Germany). For supporting the high-pressure hydraulic pistons, a custom steel frame is mounted on each side of the press, as noted in Figure 3b. The input data system is composed of a hydraulic pressure sensor and a video feed provided by a USB camera that is mounted on a FOBA ASLAI tripod, shown in Figure 4a; the camera and the hydraulic press are not in direct contact, avoiding transmitting vibrations from the machine to the video feed, as the image must be still. This industrial equipment controls the displacement and speed in the vertical direction of a 150 mm radius steel die, as indicated in Figure 4b. The hydraulic pistons are equipped with knurled gripping jaws that mechanically fix the metal sheet into place (Figure 4c). Arduino Mega microcontrollers assure the interface between the ASPF algorithm and the industrial equipment.

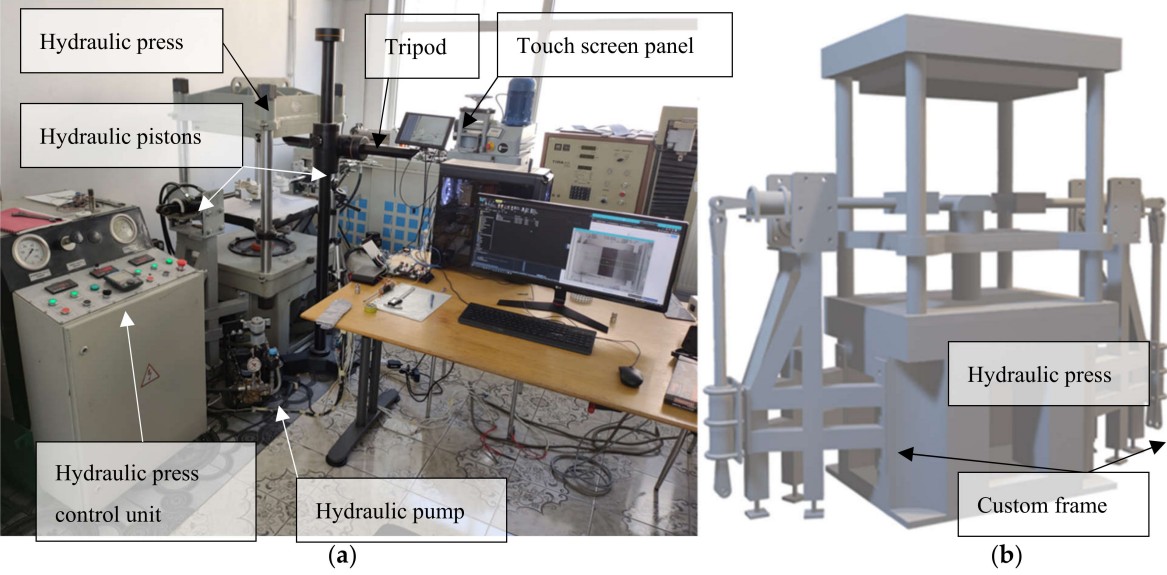

**Figure 3.** (**a**) Overview of the experimental setup including hydraulic press, hydraulic pump, data acquisition equipment; (**b**) 3D representation of the hydraulic stretch-forming press.

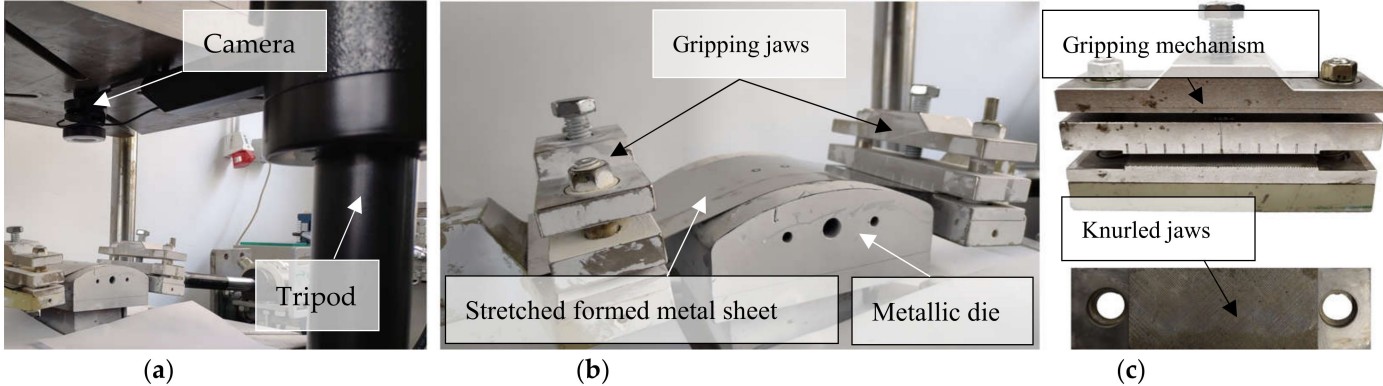

**Figure 4.** (**a**) Close-up of the experimental setup, highlighting the camera-tripod system, (**b**) steel die, stretched formed AA1050 metal sheet and (**c**) gripping mechanism assembly with knurled jaws.

The control of the hydraulic pistons in the horizontal direction is made through pressure adjustment; in this experimental study the angular displacement is not restrained, as this setup allows angular control, as noticed in Figure 5a,b. The user interface, presented in Figure 5c,d, displays the live video-feed, with the indicated additional information, regardless of the operating mode.

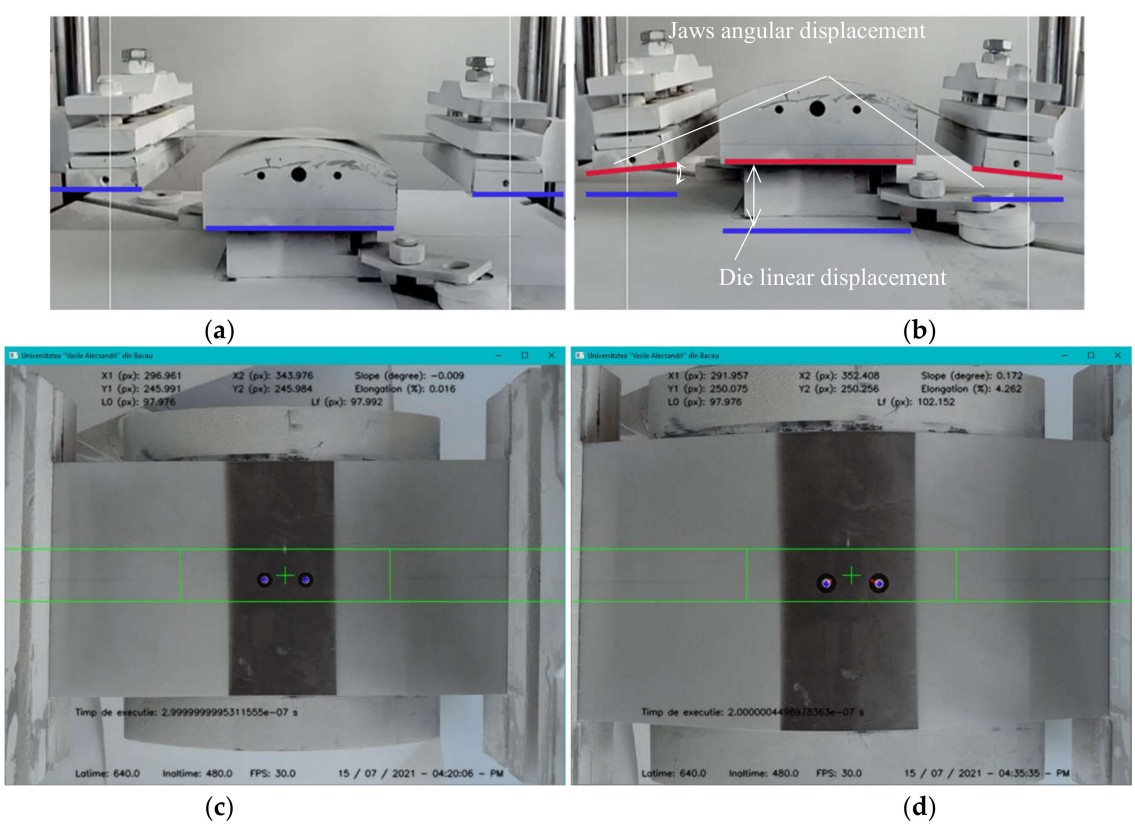

**Figure 5.** (**a**) Initial position of the stretch-forming setup; (**b**) intermediate stage of the stretch-forming process indicating the die and jaws displacement; (**c**) real-time processed video-feed at the start of the process; (**d**) real-time processed video-feed at an intermediate position.

In order to ensure an easy and efficient setup, with in-process control, a separate touch screen display was connected on which a custom Python script runs, as presented in Figure 6a; this controls the on-off status of the hydraulic pumps, piston pressure, and die speed in the vertical direction, while displaying the video-feed from the top or side camera (mounted only for setup purposes). The process repeatability represented an essential

aspect of this research study; in Figure 6b a dedicated script module interface is presented; it ensures that every part of the process (jaws, die, metal sheet, and markers) maintains its position with each run. When the ASFP algorithm starts the video-feed, it shows information about the position of each marker, the slope between markers, elongation, last decision execution time, actual time, video frame resolution, graphs for the coefficient of determination ($R^2$), and *p*-value for each decision, as indicated in Figure 6c.

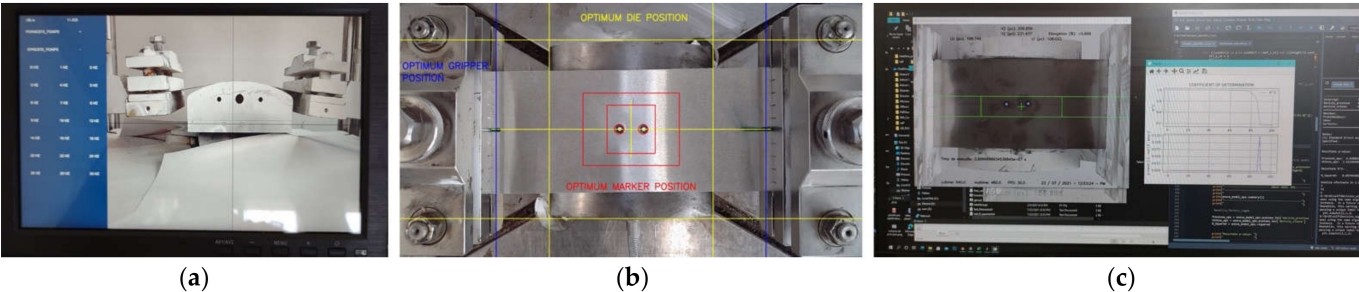

|  |  |  |
|:---:|:---:|:---:|
| (**a**) | (**b**) | (**c**) |

**Figure 6.** (**a**) Setup and in-process touch screen user interface module, displaying the control panel and side view of the process; (**b**) optimum die, gripping jaws, and marker position video-feed interface; (**c**) the ASFP algorithm user interface, showing the main live video feed along with graphs for the statistical analysis data ($R^2$ and *p*-value for each decision).

AA1050-O aluminum alloy (ALRO Vimetco, Slatina, Romania) was used for this experimental study. Considering the experimental setup arrangement of the hydraulic equipment, $100 \times 320 \times 0.5$ mm blank sheets were used, as indicated in Figure 7a. In Figure 7b, there are indicated the areas of interest that were measured and analyzed for the stretched-formed parts (material strain, part radius, and part height). The shape of the stretched parts is relative to the die shape, shown in Figure 7c,d. The analysis was conducted to evaluate our algorithm results, comparing them to those of a classical, programmed method.

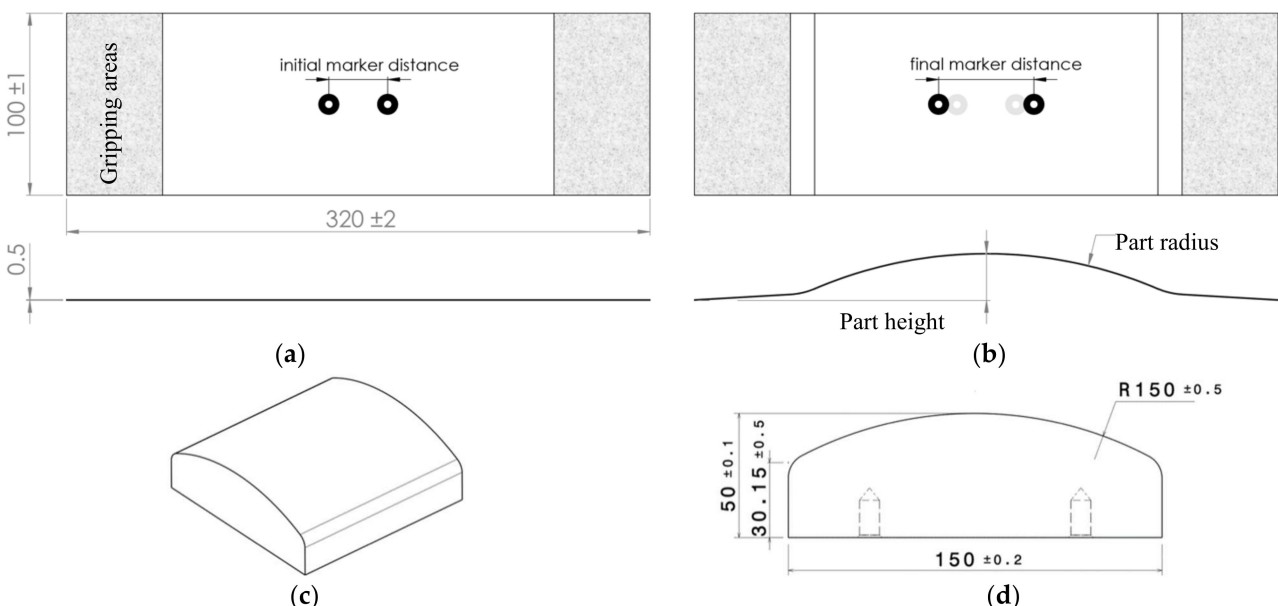

**Figure 7.** (**a**) Blank material dimensions, highlighting the gripping areas and initial marker position; (**b**) stretch-forming measured parameters (marker relative position, part radius, part height); (**c**) isometric view of the metallic die; (**d**) metallic die overall dimensions.

The chemical alloy composition, as indicated in the ASM [48], is highlighted in Table 2. The mechanical properties for this alloy were obtained on a Lloyd EZ50 material tensile machine (Lloyd Instruments Ltd, Bognor Regis, United Kingdom), using an Epsilon-3542 axial

extensometer (Epsilon Technology Corp, Wyoming, United States of America). Despite the low values for the yield strength ($R_{p0.2}$ = 28 MPa) and UTS ($R_m$ = 76 Mpa), this alloy was used for the specific state "O", having improved elongation at break capability; compared to other states (H14, H16, or H18) that offer superior yield and ultimate tensile strength, but only 8 to 12% elongation, this alloy was capable of up to 39% elongation at break. It was the subject of numerous studies, including the analysis of corrosion resistance, mechanical properties, and microstructure by continuous closed die forging [49] or carbonized eggshell [48], hybrid surface nanocomposite by multi-pass friction stir processing [50], thermal stability after equal channel angular pressing [51], layered sandwich composite materials [37,38], and explosive welded laminate [52].

**Table 2.** Chemical composition of the AA1050-O aluminum alloy [53].

| Chemical Composition wt.% | | | | | | | | |
|---|---|---|---|---|---|---|---|---|
| Al | Cu | Fe | Mg | Mn | Si | Ti | V | Zn |
| >99.5 | <0.05 | <0.4 | <0.05 | <0.05 | <0.25 | <0.03 | <0.05 | <0.05 |

A GOM Atos II 400 3D scanner image measurement system, highlighted in Figure 8a, was used to obtain precise measurements. Before spraying with the MR Chemie 2000 L anti-reflexive coating, 3/7 mm GOM markers were attached to each part. The measurements were carried out in pairs of two, side by side, with a narrow gap between parts; this was necessary for boundary identification and comparison purposes. The parts were placed on a rotating device in order for measurements from multiple angles to be made; between 7 and 12 measurements were taken for each set of parts so a continuous mesh could be generated, as show in Figure 8b. The height of each part is relative to the same measurement plane and represented the maximum distance indicated by the measuring software (Figure 8d). However, in order to measure the radius for each part, two perpendicular planes were created, intersecting at the maximum height point. A median intersection plane was created in the direction of the length of each mesh, and two other parallel planes were offset on each side at 25 mm each. In the transversal direction from the main plane, the other two planes were offset at 50 mm each. A total number of nine intersection points were created, three on each of the longitudinal planes; in this direction, three-point circles were created, as highlighted in Figure 8c, and for each part the value of the radius was obtained as a mean of the radius of these circles. Further reference will be made to these values as the average part radius for both the programmed (APR-P) and the ASFP (APR-ASFP) obtained parts.

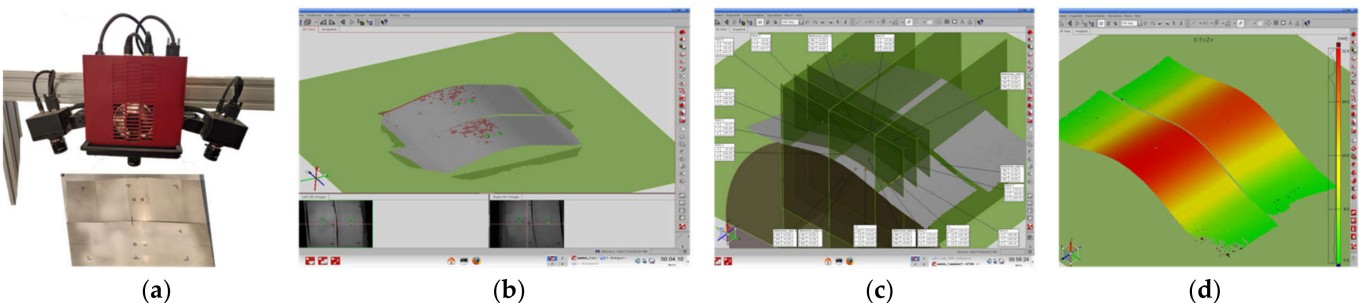

(**a**)        (**b**)        (**c**)        (**d**)

**Figure 8.** (**a**) GOM Atos II 400 3D scanner image measurement system; (**b**) side by side 3D mesh of the scanned parts obtained by the programmed and the ASFP processes; (**c**) average radius measurement points at the intersection planes; (**d**) measurement distribution layout across the 3D mesh.

A comparison of the ASFP process to a standard, programmed process was used to validate the algorithm. Furthermore, the programmed process was conducted by applying a design of experiments (DOE) by analyzing, conducting, evaluating, and interpreting the obtained data [54], with the Design-Expert software, which was necessary to be a validated

experimental plan for evaluating the ASFP algorithm. Therefore, the data obtained from the programmed process was assessed by using an analysis of variance (ANOVA), considering the effects that the process factors (stretching pressure, die speed, and die stroke) have on the responses (strain, part radius, and part height). The purpose was to determine if the process was significant to validate our statistical analysis algorithm considering the statistically validated data.

## 3. Results

A general direction emerged from the statistical analysis of the programmed process, as this highlighted what path our algorithm should follow. Therefore, after conducting the programmed process and having the maximum achieved strain for each part, the most logical decision was to declare this as a goal in our algorithm; this allowed us to compare the results to our method. Once the program was running, it decided what was best for each step to achieve the indicated strain limit.

### 3.1. Statistical Analysis Programming Process

As mentioned previously, in order to determine if this self-decision method was reliable, a comparative study was performed between the programmed and the ASFP processes. For the programmed process, an experimental plan was generated as part of a statistical study by using the response surface methodology and by applying an ANOVA analysis; a total number of 20 samples were suggested to be tested with variations of the stretching pressure, die speed, and stroke, as indicated below in Table 3. The hydraulic press speed in the vertical direction was controlled by changing the motor frequency in Hz; the main panel also displayed the speed in mm/s. As the die speed can only be controlled by changing the motor frequency, the ANOVA analysis was conducted accordingly; this was also available when the ASFP algorithm controlled the hydraulic press. The correlation between the die control frequency in Hz and the die speed in mm/s is presented in Table 3 for a complete insight into the process.

**Table 3.** The plan of the design of the experiments indicating the programmed stretch-formed process factor values, the response results, and the correlation between the die control frequency in Hz and the die speed in mm/s, sorted in ascending order by strain.

| Part. Number | Stretching Pressure Bar | Die Control Frequency Hz | Die Speed mm/s | Die Stroke, Programmed mm | Strain % |
|---|---|---|---|---|---|
| 13 | 0 | 30 | 2 | 30 | 6.75 |
| 10 | 0 | 1 | 0.03 | 37.4 | 7.43 |
| 9 | 11.8 | 2 | 0.06 | 38.3 | 8.26 |
| 16 | 0 | 19 | 1.68 | 50 | 8.42 |
| 7 | 0 | 1 | 0.03 | 50 | 8.82 |
| 17 | 9 | 14 | 1.08 | 30.4 | 8.85 |
| 12 | 9 | 14 | 1.08 | 30.4 | 9.05 |
| 11 | 9 | 29 | 1.84 | 41 | 9.10 |
| 4 | 9 | 29 | 1.84 | 41 | 9.20 |
| 6 | 10 | 28 | 1.84 | 30.9 | 9.75 |
| 15 | 9 | 29 | 1.84 | 41 | 9.80 |

**Table 3.** *Cont.*

| Part. Number | Stretching Pressure Bar | Die Control Frequency Hz | Die Speed mm/s | Die Stroke, Programmed mm | Strain % |
|---|---|---|---|---|---|
| 19 | 3.2 | 18 | 1.56 | 38.3 | 9.98 |
| 18 | 19.6 | 14 | 1.08 | 41 | 10.09 |
| 8 | 19.6 | 14 | 1.08 | 41 | 10.45 |
| 1 | 19.6 | 14 | 1.08 | 41 | 10.53 |
| 5 | 20 | 1 | 0.03 | 30 | 11.2 |
| 3 | 11.7 | 18 | 1.56 | 50 | 11.92 |
| 2 | 20 | 30 | 2 | 30 | 12.5 |
| 14 | 20 | 30 | 2 | 50 | 12.58 |
| 20 | 12.5 | 1 | 0.03 | 50 | 12.75 |

The ANOVA results, presented in Table 4, indicated that the study was significant with values for each response for the $R^2$ exceeding 0.9, while the difference between the adjusted and predicted $R^2$ was less than 0.2; the Adequate Precision, as signal to noise ratio, was above the significant value of 4. Taking this into account, the programmed experimental study was a solid base for a suggestive comparison with the ASFP.

**Table 4.** Fit statistics of the ANOVA analysis for the programmed stretch-forming process, for each response.

| Fit Statistics | Strain | Part Radius | Part Height |
|---|---|---|---|
| $R^2$ | 0.9888 | 0.9219 | 0.9504 |
| Adjusted $R^2$ | 0.9735 | 0.8352 | 0.9215 |
| Predicted $R^2$ | 0.8432 | 0.7426 | 0.7933 |
| Adequate Precision | 28.1992 | 9.7440 | 17.5590 |

Furthermore, in order to understand the behavior of the deformed material regarding the responses, a complete analysis was conducted. It was essential to predict how each factor influenced the responses and to what extent.

The analysis of the strain model returned a *p*-value below 0.0001, which indicated that it was highly significant. Moreover, the die stroke was the most relevant factor in obtaining higher values, as expected; this relation was shown in Figure 9a, where the analysis predicted a maximum strain of 16.25% if the pressure was 20 bar, the die speed was at 0.03 mm/s with a stroke of 50 mm. This was relevant in our case as it gave insight into how our algorithm should behave. Furthermore, the statistical data inferred that the stretching pressure, in combination with the die speed, had the same effect on the strain. As a result, it can be noted from Figure 9b,c that the strain limit values were obtained not only for the maximum value of the factors, but also for the middle ranges.

In the case of the part radius, the model was significant with a *p*-value of 0.008. The interaction between the stretching pressure and the die stroke significantly impacted on obtaining the desired radius, as shown in Figure 10a. In this case, a large die stroke led to lower values of the radius. Conversely, the analysis indicated high stroke values resulted in parts with a radius in the range of 150 mm. With a *p*-value lower than 0.05, the analysis revealed that the interaction of all factors could lead to attaining the desired values of the radius. It can be noted from Figure 10a–c that the most suggestive interaction occurred using a 50 mm stroke, combined with a speed below 0.24 mm/s and a stretching pressure above 15 bar.

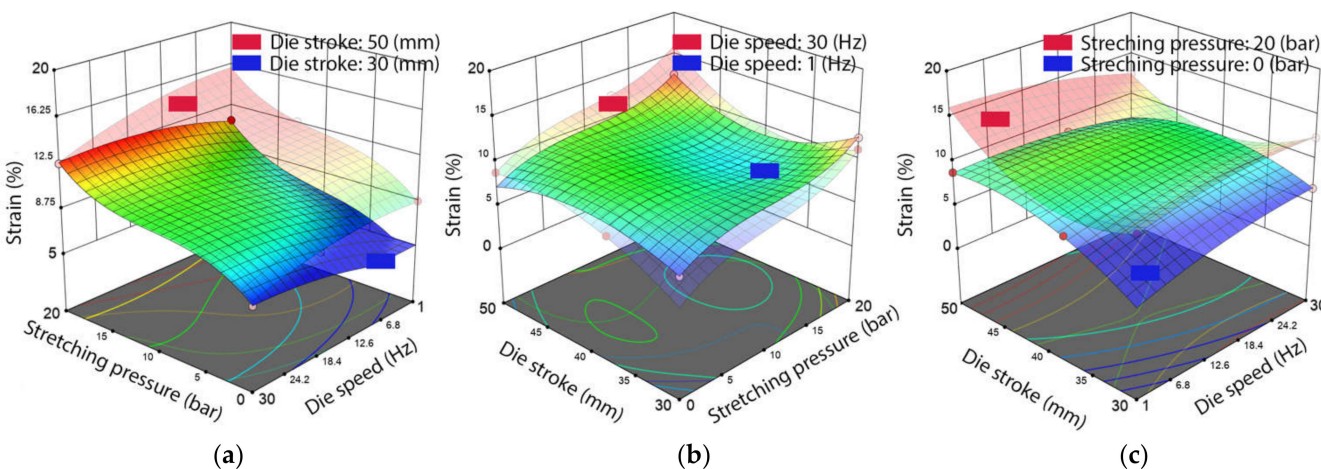

**Figure 9.** Results of the ANOVA analysis highlight the influence of the process parameters on the material strain, indicating lower and upper limits for (**a**) die stroke, (**b**) die speed and (**c**) stretching pressure.

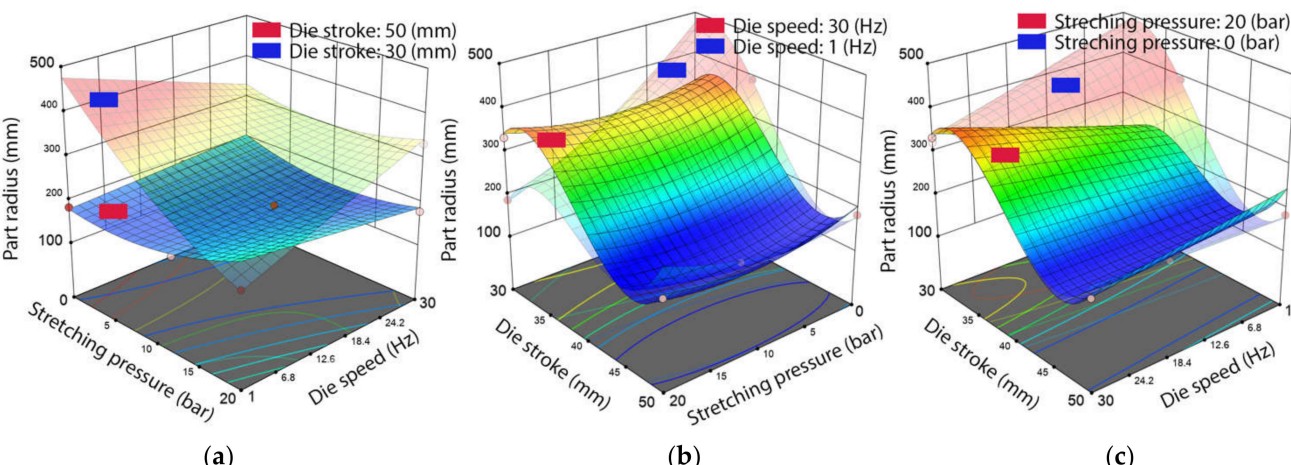

**Figure 10.** Results of the ANOVA analysis highlight the influence of the process parameters on the part radius, indicating lower and upper limits for (**a**) die stroke, (**b**) die speed and (**c**) stretching pressure.

In the case of the part height, the model was considered significant, with a calculated *p*-value less than 0.0001. The interaction between stretching pressure and die stroke, along with the die speed were relevant in this case. Figure 11a,b indicated that a higher value of the part height was related to higher die strokes; the stretching pressure had a significant impact when using lower die strokes, acting as a compensation factor. The variation in the die speed did not lead to significant adjustments to the part height.

The industrial nature of the SFP implied obtaining parts with a radius as close as possible to that of the die. As indicated by the ANOVA analysis, this could be achieved when using higher die strokes and stretching pressures. This combination of factors can be unsafe, due to excessive strain and stress, leading to material failure. It was necessary to specify that lower speeds were suggested for optimal results; this significantly interfered with any industrial process, as the production time increased. An automated approach may be a viable solution, so that the maximum strain and optimal radius may be obtained without compromising the process speed.

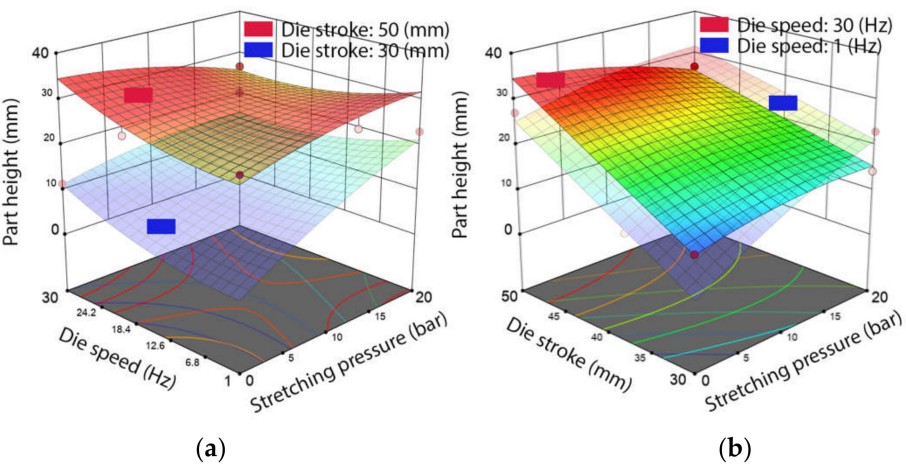

**Figure 11.** Results of the ANOVA analysis highlighting the influence of the process parameters on the part height, indicating lower and upper limits for (**a**) die stroke and (**b**) die speed.

### 3.2. Overall Results for the Adaptive Stretch-Forming Process Algorithm

The complete overview of the stretch-formed parts is presented in Figure 12a,b. An insight on the deformability improvement emerged from the early stages, from the first self-stretched parts. Furthermore, the side-by-side GOM measurement indicated this improvement. At this stage, the data regarding strain for each part were recorded and used as limits for the ASFP algorithm. The purpose was to compare the obtained parts for the degree of deformation as deviation from the die radius, part height, maximum die stroke, and process time. A simple method was used to compare the parts: the die radius was divided by the average part radius (APR), obtaining a deformation coefficient; if the coefficient was 1, the part radius was equal to the die radius. The complete data are presented in Table 5. The APR measured values and the determined deformation coefficients were presented along with the part height for both the programmed and the ASFP parts. The ASFP algorithm recorded the data regarding each process in .csv files; there was a process time among the readings. The programmed process time values were recorded for the constant die speed and the predetermined stroke, while those for the ASFP were dependent on a programmed goal (maximum strain), therefore it varied with each decision (variable die speed, stretching pressure).

**Table 5.** Results of the programmed and ASFP for average part radius, height, process time, and deformation coefficient, sorted in ascending order by strain.

| Part. Number | APR-P mm | APR-ASFP mm | Die Radius /APR-P Coefficient | Die Radius/APRA-SFP Coefficient | Part Height, Programmed mm | Part Height, ASFP mm | Process Time, Programmed s | Process Time, ASFP s |
|---|---|---|---|---|---|---|---|---|
| 13 | 321.43 | 163.25 | 0.4667 | 0.9188 | 11.5 | 27 | 15 | 132 |
| 10 | 391.26 | 184.44 | 0.3834 | 0.8133 | 9.98 | 26.2 | 1246.7 | 257 |
| 9 | 210.35 | 154.37 | 0.7131 | 0.9717 | 20.2 | 29.8 | 638.4 | 223 |
| 16 | 187.71 | 154.53 | 0.7991 | 0.9707 | 26.9 | 24.7 | 29.8 | 189 |
| 7 | 183.48 | 160.96 | 0.8175 | 0.9319 | 27.2 | 26.5 | 1666.7 | 334 |
| 17 | 321.02 | 184.39 | 0.4673 | 0.8135 | 13 | 26.1 | 28.2 | 215 |
| 12 | 323.7 | 177.34 | 0.4634 | 0.8458 | 12 | 27.1 | 28.2 | 169 |
| 11 | 173.95 | 150.04 | 0.8623 | 0.9997 | 24.7 | 24.3 | 22.3 | 232 |
| 4 | 178.18 | 190.21 | 0.8418 | 0.7886 | 24.6 | 26.4 | 22.3 | 205 |
| 6 | 337.43 | 174.39 | 0.4445 | 0.8601 | 11.9 | 29 | 16.8 | 334 |
| 15 | 202.46 | 159.47 | 0.7409 | 0.9406 | 25.8 | 30.2 | 22.3 | 358 |
| 19 | 314.85 | 189.96 | 0.4764 | 0.7896 | 16.6 | 23.1 | 24.6 | 223 |
| 18 | 160.03 | 152.77 | 0.9373 | 0.9819 | 22.2 | 34.2 | 38 | 351 |

**Table 5.** *Cont.*

| Part. Number | APR-P mm | APR-ASFP mm | Die Radius /APR-P Coefficient | Die Radius/APRA-SFP Coefficient | Part Height, Programmed mm | Part Height, ASFP mm | Process Time, Programmed s | Process Time, ASFP s |
|---|---|---|---|---|---|---|---|---|
| 8 | 194.7 | 158.15 | 0.7704 | 0.9485 | 22.3 | 34.2 | 38 | 302 |
| 1 | 255.02 | 179.84 | 0.5882 | 0.8341 | 21.2 | 24.6 | 38 | 217 |
| 5 | 186.43 | 176.66 | 0.8046 | 0.8491 | 23.1 | 36.2 | 1000 | 289 |
| 3 | 190.64 | 169.52 | 0.7868 | 0.8849 | 29.7 | 34.7 | 32.1 | 307 |
| 2 | 333.6 | 152.56 | 0.4496 | 0.9832 | 14.3 | 31.7 | 15 | 307 |
| 14 | 178.71 | 151.1 | 0.8393 | 0.9927 | 26.8 | 32.2 | 25 | 195 |
| 20 | 185.47 | 173.41 | 0.8088 | 0.8650 | 28.3 | 32 | 1666.7 | 266 |

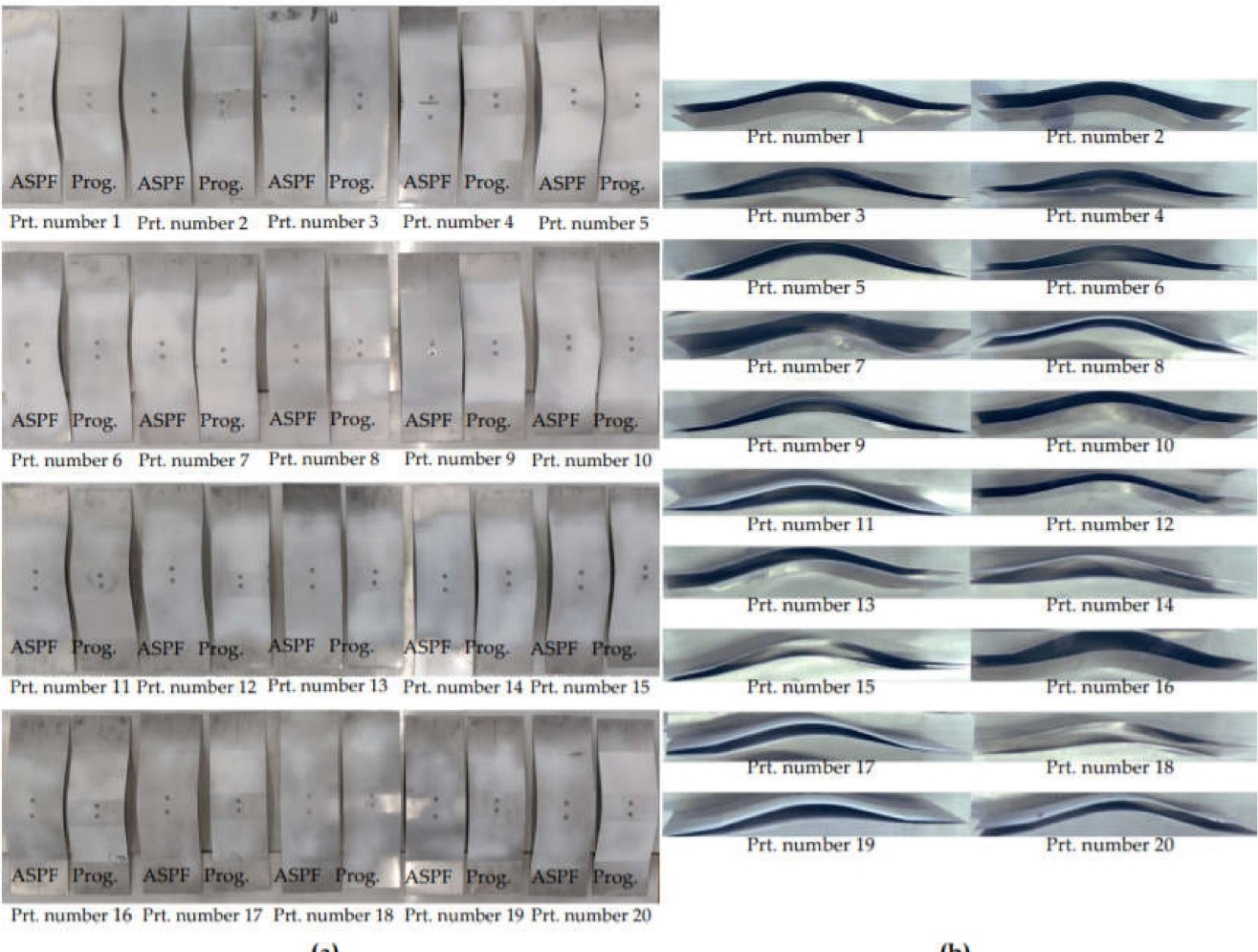

**Figure 12.** (**a**) Stretch-formed part shape comparison top and (**b**) side-view (ASFP parts on top), sorted by part number.

## 4. Discussion

On average, the deformation coefficient for the programmed process is 0.673; when using the ASFP, this value increases to 0.899, representing an improvement of 34% in deformability. The parts can now be analyzed comparatively. For this experimental study to validate our algorithm, the following cases are presented:

- best-case match for part-to-die by radius (Table 6);
- worst-case match for part-to-die by radius (Table 7);
- best and worst case matches for part-to-part by radius for both the programmed and self-decision algorithm (Table 8).

Table 6 highlights the results in the range of the die radius as a comparison between the programmed and the ASFP parts. Furthermore, the results are presented as a comparison of each part obtained by each method. The best match for part-to-die by radius was obtained for the programmed process at part number 18 and the ASFP at part number 11. When the stretch-forming uses the programmed method for part number 18, the radius has a value of 160.03 mm, resulting in a deformation coefficient of 0.9373. In contrast, part number 11 has a radius of 150.04 mm, with a deformation coefficient of 0.9997. This difference was obtained by a reduction in strain from 10.09% to 9.1% and an increase in part height of 2.1 mm. The differences are noticeable when comparing the result obtained for the same part. For the same strain value of 10.09%, the ASFP offers a deformation coefficient of 0.9819 at a radius of 152.77 mm, with a part height of 34.2 mm. Improvements are observed in the case of part number 11, with an increase in the deformation coefficient from 0.8623 to 0.9997.

**Table 6.** Best-case match for die-to-part by radius with measurements for the programmed and the ASFP parts.

| Match by Radius | Part Number | GOM Measurement Comparison | |
| --- | --- | --- | --- |
| | | Programmed | ASFP |
| Best-case match part-die | 18 | #18 prog<br>Strain: 10.09 (%)<br>Height: 22.2 (mm)<br>Radius: 160.03 (mm) | #18 ASFP<br>Strain: 10.09 (%)<br>Height: 34.2 (mm)<br>Radius: 152.77 (mm) |
| | 11 | #11 prog<br>Strain: 9.1 (%)<br>Height: 24.7 (mm)<br>Radius: 173.95 (mm) | #11 ASFP<br>Strain: 9.1 (%)<br>Height: 24.3 (mm)<br>Radius: 150.04 (mm) |

Recording the process parameters for both methods highlighted the dynamics of each process in terms of the number of decisions and what they represent for the stretching pressure, die speed, stroke, strain, process $R^2$, and process total time, as presented in Figure 13 (for part number 18) and in Figure 14 (for part number 11). While the programmed process was constant for both pressure and die speed, the ASFP process indicated that the algorithm adapted as the metal sheets deformed. In the case of part number 18, 3.427 [decisions/second] were taken; the ASFP algorithm managed to keep the $R^2$ value above the 0.8 thresholds for 250 decisions, until the strain value reached 1.5%. A particular aspect emerged from the complete data analysis, as in the $R^2$ value; a sudden drop occurred after a constant slope. This area could be assigned as the transition from the elastic to plastic deformation. This prediction correlated to the algorithm increasing the stretching pressure and the die speed as a compensation mechanism. Considering that the elastic area implied constant stress to the strain slope, the algorithm's statistical analysis found it relevant, and the process model was mathematically predictable. Once this boundary was crossed, the algorithm was confronted with a model that was not as predictable. Depending on each case, it tried to find the correct speed and pressure to raise the value of the $R^2$. When analyzing the blue areas of the graph, the zones with the same speeds have a color fill area, while those with different attempts appear faded.

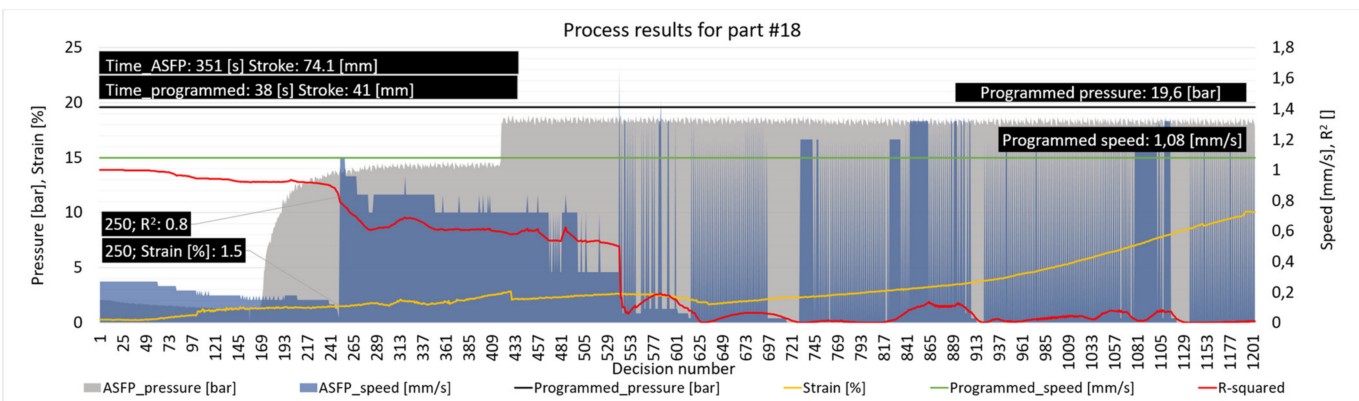

**Figure 13.** Programmed and ASFP parameters, indicating stretching pressure, die speed, die stroke, material strain, process $R^2$, number of decisions, and total process time for part number 18.

The algorithm decreased the speed or stops of the die displacement, while it analyzed its decision to find if it was relevant or not. It seemed that the stretching pressure was considered the most relevant factor, and accordingly, its value was slightly modified. Nevertheless, the unpredictable behavior of the material, while stretching beyond the elastic area, did not represent an obstruction for generating and evaluating decisions.

The ASFP lasted for 351 s when stretch-forming part number 18; compared to the programmed process that lasted for 38 s, an increase of 6% in the deformation coefficient was obtained. It did not represent the difference between the process times, but if considered, this part was among the closest to the die shape. It was not a general rule, as presented in the following cases.

When analyzing the decision distribution for part number 11, it was noted that while the $R^2$ value dropped much faster, after 70 decisions, the algorithm tried to correct this by changing the die speed. In this case, the strain was 0.32% when the $R^2$ value was 0.8. In contrast with the decision made regarding part number 18, the pressure was considered relevant after 150 decisions, with a sudden and massive increase approximatively starting with the decision number 400. Within this interval, the growth was slow with variations of the die speed. Compared to part number 18, this run was faster, with a total time of 232 s; a decrease of 33.9% was recorded for a reduction in the strain of 9.81% when comparing the two parts.

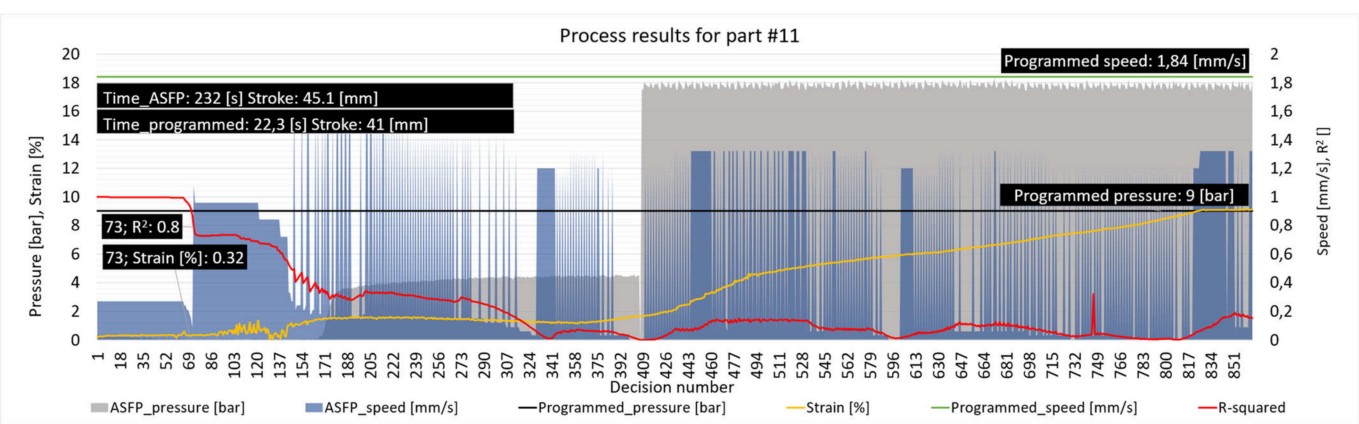

**Figure 14.** Programmed and ASFP parameters, indicating stretching pressure, die speed, die stroke, material strain, process $R^2$, number of decisions, and total process time for part number 11.

The second set of comparisons highlighted the worst-case match between part and die by radius. The data was presented as previously described in Table 7. Part number 10,

stretch-formed using the programmed process, resulted in a radius of 391.26 mm, with a deformation coefficient of only 0.3834 and a height of 9.98 mm.

**Table 7.** Worst-case match for die-to-part by radius with measurements for the programmed and the ASFP parts.

| Match by Radius | Part Number | GOM Measurement Comparison | |
|---|---|---|---|
| | | **Programmed** | **ASFP** |
| Worst-case match part-die | 10 | #10 prog Strain: 7.43 (%) Height: 9.98 (mm) Radius: 391.26 (mm) | #10 ASFP Strain: 7.43 (%) Height: 26.2 (mm) Radius: 184.44 (mm) |
| | 4 | #4 prog Strain: 9.2 (%) Height: 24.6 (mm) Radius: 178.18 (mm) | #4 ASFP Strain: 9.2 (%) Height: 26.4 (mm) Radius: 190.21 (mm) |

Compared to the process obtained by using the ASFP, huge differences were noted with 112% improvement in the deformation coefficient, to a value of 0.8133 and a part height of 26.2 mm; this agreed with the data presented in Figure 15, where the process was mainly conducted at pressures below 12.5 bar. The algorithm maintained the R2 value above the 0.8 thresholds for 154 decisions until reaching a strain of 0.26%. When analyzing the speed values after the decision 205, when the R2 dropped significantly, it was noted that the algorithm considered that frequent stops of the die movement were necessary. The result further confirmed the conclusions drawn from the statistical analysis of the programmed process, where middle to high pressure was indicated for optimal values.

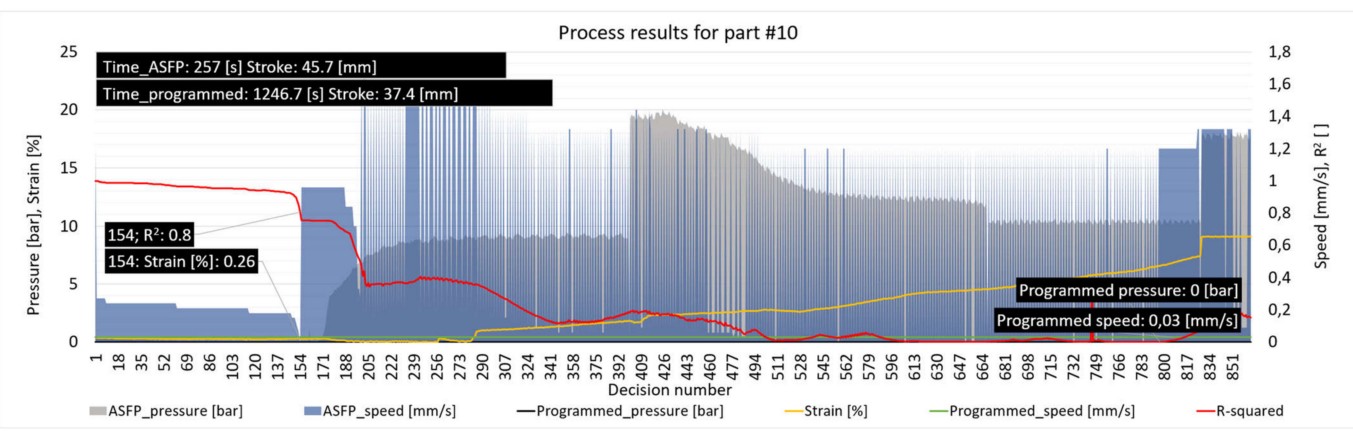

**Figure 15.** The programmed and the ASFP parameters, indicating the stretching pressure, the die speed, the die stroke, the material strain, the $R^2$ process, the number of decisions, and the total process time for part number 10.

When analyzing the worst-case match obtained by the ASFP (part number 4), it was noticed that the radius decreased by 3.13%, in comparison with part number 10. Particularities emerged as the ASFP was at a disadvantage, compared to the programmed process as the radius increased by 7% from 178.18 to 190.21 mm.

In this case, the decisions were similar to those made for the other parts, as resulting from Figure 16, where it was noted that the coefficient of determination dropped below 0.8 after 170 decisions; consequently, the strain value was 0.32%. A decrease in the $R^2$

value was evaluated as the increase in pressure and in successive die speed adjustments. The attempts to return to a higher value were spread throughout 250 decisions, until the decision 426, when the pressure was increased up to 18 bar. From this point until the end of the process, only the die speed was intensively controlled.

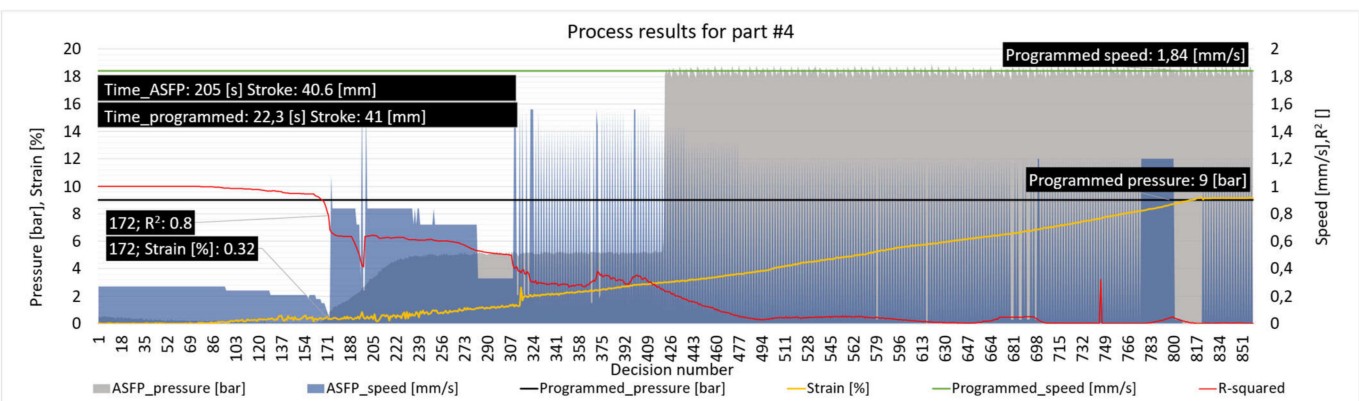

**Figure 16.** Programmed and ASFP parameters, indicating stretching pressure, die speed, die stroke, material strain, process $R^2$, number of decisions, and total process time for part number 4.

Furthermore, a precise distinction was evident when comparing part number 10 with part number 4; even the worst-case match offered by the ASFP was much closer to the die radius than the programmed part was.

The data that resulted from comparing the parts with the same strain obtained by each method was presented in Table 8. In the case of part number 5, the measurements indicated a difference of 6% in the deformation coefficient, with the ASPF offering the best results; this improvement was noted in the part height and in the processing time; taking into consideration that the programmed process lasted for 1000 s compared to 289 s for the ASFP, as indicated in Figure 17.

**Table 8.** Best and worst-case match by part-part radius with measurements for programmed and ASFP parts.

| Match by Radius | Part Number | GOM Measurement Comparison | |
|---|---|---|---|
| | | **Programmed** | **ASFP** |
| Best-case match part-part | 5 | #5 prog<br>Strain: 11.2 (%)<br>Height: 23.1 (mm)<br>Radius: 186.43 (mm) | #5 ASFP<br>Strain: 11.2 (%)<br>Height: 36.2 (mm)<br>Radius: 176.66 (mm) |
| Worst-case match part-part | 2 | #2 prog<br>Strain: 12.5 (%)<br>Height: 14.3 (mm)<br>Radius: 333.6 (mm) | #2 ASFP<br>Strain: 12.5 (%)<br>Height: 31.7 (mm)<br>Radius: 152.56 (mm) |

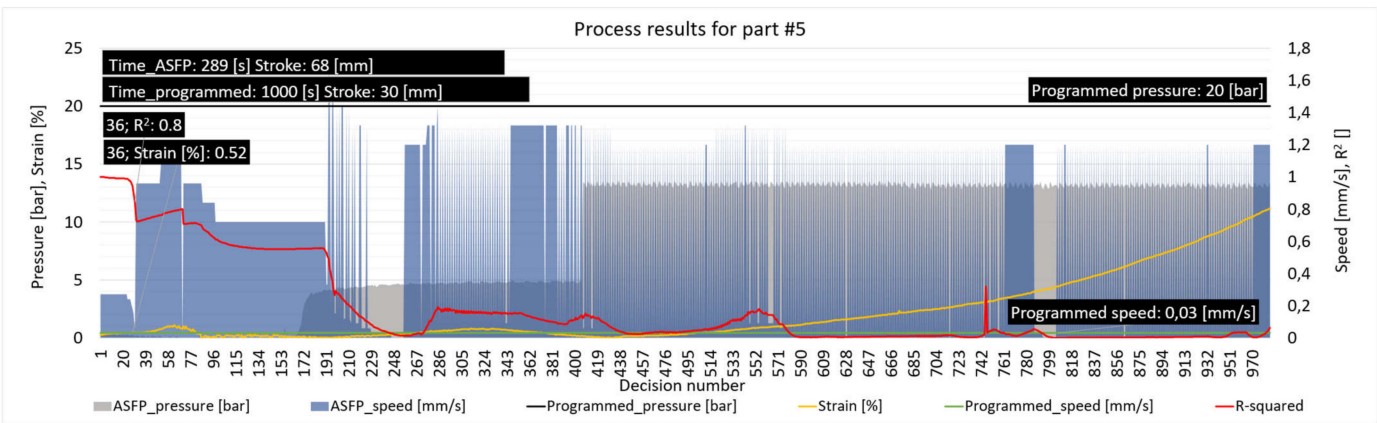

**Figure 17.** Programmed and ASFP parameters, indicating stretching pressure, die speed, die stroke, material strain, process R2, number of decisions, and total process time for part number 5.

When comparing the overall process parameters, it was noted that the programmed process was a resource consumer, running at low speed with a pressure of 20 bar, in contrast to the ASFP that used higher die speeds with a gradual increase in pressure, while not exceeding 13 bar. A strain of 0.52% occurred after only 36 decisions; this was assigned to the fact that the process started with a die speed above 0.3 mm/s, which was a particular case.

The most notable difference in the part-to-part match by radius, highlighted in Figure 18, was obtained for part number 2. A radius of 333.6 mm was obtained by using the programmed process, while the ASFP managed to process the part to a radius of 152.56 mm. The difference in the deformation coefficient was from 0.4496 to 0.9832, with a percentage difference between parts radius of 119%. Furthermore, the height of the ASFP part was 31.7 mm, 112% higher. Compared to the other parts, in this case, the $R^2$ value dropped sharply after 85 decisions while the stretching pressure was turned off. This behavior had not been observed in other cases, since no pressure was used for almost 150 decisions, provided that the algorithm tried in every case to raise the stretching pressure when the $R^2$ value dropped quickly.

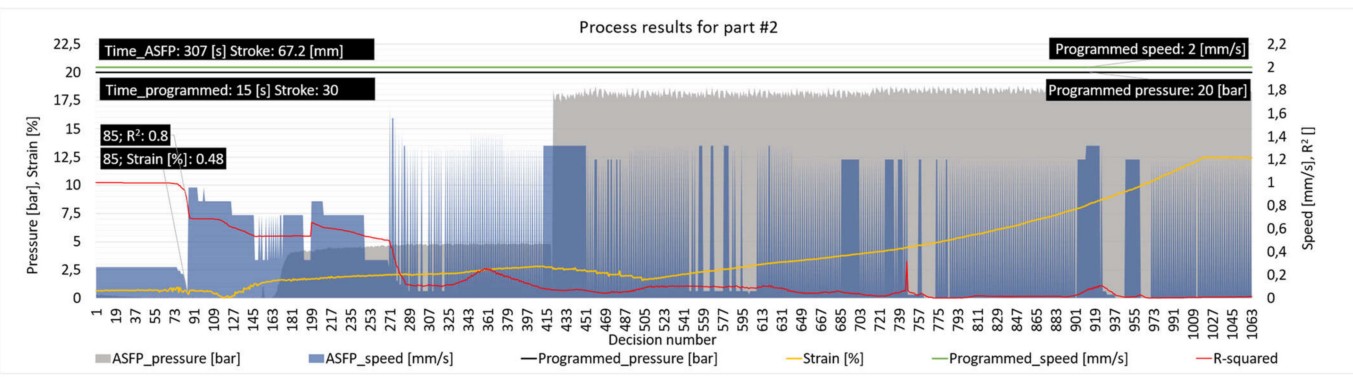

**Figure 18.** Programmed and ASFP parameters, indicating stretching pressure, die speed and die stroke, material strain, process $R^2$, number of decisions, and total process time for part number 2.

The data related to the dimensional accuracy (part radius and height) of the ASFP highlights this method's improvements. The analysis of the algorithm includes the review of the processing time and the number of decisions. As indicated in Figure 19, the ASFP is 23% faster, with an average process time of 255.2 s. The processing time ranges from 132 to 358 s, while the programmed method ranges from 15 to 1666.7 s.

The number of decisions ranged from 549 to 1203, meaning an average value of 887.4. The frequency at which the algorithm made and implemented its decisions was between 2.156 and 4.689 decisions/second, an average of 3.477 decisions/second. The frequency

proved to be optimum, as some time was needed for the hydraulic equipment to perform the commands. From the point of view of the statistical analysis calculation time, no impediment had emerged, as one complete calculation lasted for $4 \times 10^{-7}$ s.

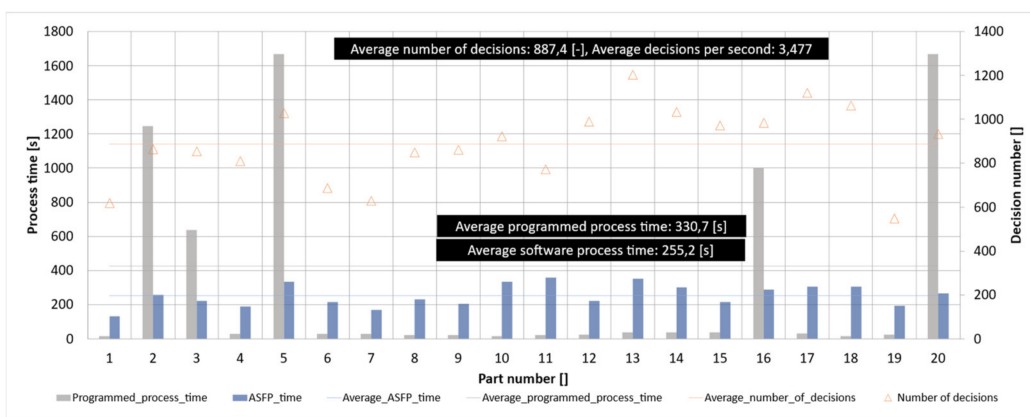

**Figure 19.** Comparison of process time between stretch-formed parts by using the programmed and the ASFP algorithm processes, number of decisions, and average number of decisions.

## 5. Conclusions

The ASFP algorithm provided a new insight into using computer vision and statistical analysis in the stretch-forming process. In this experimental study, a comparison between a classical method and our algorithm was conducted, highlighting the improvements in deformability and process time. The results indicated that the deformation coefficient was improved by 34%, and the overall average process time decreased by 23%.

Furthermore, the experimental data obtained from the programmed stretch-forming process offered a statistical interpretation of the effects of the process parameters (stretching pressure, die speed, and die stroke) on the material strain, part radius, and part height.

Being in a continuous stage of development, this technique had the benefit of constant improvement from the data analysis of different alloys and stretch-forming conditions. The system can import real-time data and establish a connection between the process parameters and strain, part radius, and part height, thus deciding and evaluating its next steps.

**Author Contributions:** Conceptualization: C.C.G. and B.C.; methodology: C.C.G., V.Z. and B.C.; software: C.C.G.; validation: V.Z., B.C. and V.A.C.; formal analysis: C.C.G.; investigation: C.C.G., B.C. and V.A.C.; resources: V.Z. and B.C.; data curation: C.C.G., B.C. and V.A.C.; writing—original draft preparation: C.C.G. and B.C.; writing—review and editing: V.Z., B.C. and V.A.C.; visualization: C.C.G.; supervision: V.Z. and B.C.; project administration: V.Z. and B.C.; funding acquisition: V.Z. All authors have read and agreed to the published version of the manuscript.

**Funding:** This research was funded by the Ministry of Education and Research, through the National Council for the Financing of Higher Education, Romania, grant number CNFIS-FDI -2021-0276, and by the Ministry of Research and Innovation, through the Executive Unit for Financing Higher Education, Research, Development and Innovation, Romania, project number PN-III-P1-1.2-PCCDI-20170446/82PCCDI/2018, within PNCDI III.

**Institutional Review Board Statement:** Not applicable.

**Informed Consent Statement:** Not applicable.

**Data Availability Statement:** Data is contained within this article.

**Acknowledgments:** The authors would like to acknowledge the support of PSAPET PROD-COM S.R.L, Bacău, Romania for manufacturing the metallic die and SC PRODACOM CONSTRUCT SRL for manufacturing the exterior metal frames.

**Conflicts of Interest:** The authors declare no conflict of interest.

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
