# Peer review of "Adaptive Stretch-Forming Process: A Computer Vision and Statistical Analysis Approach"

_machines, doi:10.3390/machines9120357_

Round 1

Reviewer 1 Report

In terms of publishing this paper in "Sensors" the sensor work is quite rudimentary.  Two points are used to measure strain.  It is now routine to optically capture full strain fields on 3-D objects.  

There might be something of great interest here and very much publishable about the adaptive A.I. piece of this, but I am not really qualified to review this, but that does not belong in Sensors, in my mind. Further, the discussion has a lot of extraneous content about topics like Severe Plastic Deformation (there is none) and too much detail on pedestrian aspects of the experimental setup.  

There may be something publishable here, but not for the Sensors audience. 

The authors should read papers in top rate journals and emulate the practice.

The biggest problem is the sensors content is very pedestrian.

Author Response

Response to Reviewer 1 Comments

Point 1: In terms of publishing this paper in "Sensors" the sensor work is quite rudimentary.  Two points are used to measure strain.  It is now routine to optically capture full strain fields on 3-D objects.  

Response 1: For our algorithm to work properly it needs to take decisions in real-time, based on the measurements it makes. In the development of the algorithm, extensive research in the field of 3D image measuring systems was made. Indeed, this system is widely used, but the full strain measurements are done by using dedicated post-processing software. These software tools do not offer the possibility of exporting data as recorded, in real-time, therefore we had to design the algorithm with computer vision capabilities.

Point 2: There might be something of great interest here and very much publishable about the adaptive A.I. piece of this, but I am not really qualified to review this, but that does not belong in Sensors, in my mind. Further, the discussion has a lot of extraneous content about topics like Severe Plastic Deformation (there is none) and too much detail on pedestrian aspects of the experimental setup.  

Response 2: We have considered that the functions implemented in the algorithm (computer vision, object detection, object tracking, statistical analysis, decision evaluation and industrial equipment control) are among the topics of Sensors journal (localization and object tracking, vision/camera based sensors, action recognition and artificial intelligence in sensing and imaging) Furthermore, the successful implementation of this algorithm to an industrial process represents, in our opinion, a step forward in developing an adaptive or assisted software solution. The severe plastic deformation nature of the process, indicated in line 27, represents an important aspect of this process and is the subject of numerous scientific research. Our goal is to present the results obtained by an adaptive system implemented for the stretch-forming process, therefore the experimental setup offers information for both the software and industrial equipment’s components.

Point 3: There may be something publishable here, but not for the Sensors audience. 

Response 3: As indicated at point 2, we consider that the implemented functions from our algorithm are among the topics/scopes of the Sensors journal.

Point 4: The authors should read papers in top rate journals and emulate the practice.

Response 4: Taking into consideration these indications, and those suggested in the other two reviews, the manuscript was revised by our English department; further changes were made by adding additional explanations, changing or improving figures.

Point 5: The biggest problem is the sensors content is very pedestrian.

Response 5: We consider that the sensor content is focused on the software nature of the manuscript, as indicated in the response given at point 2.

Reviewer 2 Report

Reading the text raised some doubts / remarks / comments that should be clarified / removed / completed:

1.Introduction

-Fig. 1 must be performed again, taking into account the marking of the plate, forming tool, etc.

2.ASFP Algorithm and Industrial Setup

- In Fig. 3a and Fig. 4 individual elements of the test stand equipment should be marked and described

-Please show the differences in the photos in Fig. 5a

-Please provide detailed information on the GOM ATOS 3D scanner and the measurement methodology

-on what are the test results presented in Table 2

-Fig.12a-please explain / mark on the photo ... (programmed on the right, ASFP on the left)

3.Results or 4.Discussion

-It seems interesting to change the thickness of the plastically shaped sheets. The results have not been presented in the publication, and the use of a 3D scanner for measurement makes it possible to obtain these values

4.Conclusions

  Avoid personal statements such as: We have shown in this paper that (Line 427) And make them impersonal throughout the publication

Author Response

Response to Reviewer 2 Comments

Point 1: Fig. 1 must be performed again, taking into account the marking of the plate, forming tool, etc.

Response 1: Figure 1 has been redesigned indicating the die, press table, hydraulic ram, gripping jaws, metal sheet blank and deformed shape, jaw control, die and stretching force.

Point 2: In Fig. 3a and Fig. 4 individual elements of the test stand equipment should be marked and described

Response 2: The figures were modified and the industrial equipment is described accordingly, indicating with text the components of the industrial experimental setup.

Point 3: Please show the differences in the photos in Fig. 5a

Response 3: Figure 5 has been replaced. Figures 5a, 5b, 5c and 5d now show a more detailed view, indicating the die linear displacement and jaw angular displacement. The description of Figure 5 was changed accordingly.

Point 4: Please provide detailed information on the GOM ATOS 3D scanner and the measurement methodology

Response 4: Information about the GOM ATOD 3D scanner and measuring methodology is included in the text (starting with line 172); the description now offers more detail on how the measurements were conducted.

Point 5: on what are the test results presented in Table 2.

Response 5: The information about the mechanical properties were obtained using the equipment in our material testing laboratory and the description is now included in the text (starting with line 158). The chemical composition is according to ASM, volume II, Properties and Selection: Nonferrous Allays and Special-Purpose Materials and is included in the text as a bibliographic reference.

Point 6: Fig.12a-please explain / mark on the photo ... (programmed on the right, ASFP on the left).

Response 6: The individual images that comprise Figure 12 were rearranged and indications about part number and type of process were added.

Point 7: It seems interesting to change the thickness of the plastically shaped sheets. The results have not been presented in the publication, and the use of a 3D scanner for measurement makes it possible to obtain these values.

Response 7: Measuring the thickness of the stretched material is possible for sure. However, the research conducted at this stage allows us to better understand and update the algorithm. We have planned further research on different aluminium (Al-2024-T0, Al-6061-0) and magnesium alloys (AZ31-B). This will offer us a better understanding of how we should program the algorithm to behave when working in the plastic domain.

Point 8: Avoid personal statements such as: We have shown in this paper that (Line 427) And make them impersonal throughout the publication

Response 8: Personal statements were modified throughout the paper, as suggested.

Reviewer 3 Report

  1. It is recommended to point out which components are in Figure 3. In doing this, it will be easier for the reader to understand the meaning that the authors want to express.
  2. Page 2: …improvements [7], [27].-> …improvements [7,27].
  3. Page 7: …composite materials [39], [40].-> …composite materials [39,40].
  4. Page 8, Table 3: Please indicate the difference between Die speed [Hz] and Die speed [mm/s]
  5. Page 10: Please doule check Figures 10 and 11. Are both “Results of the ANOVA analysis highlighting the influence of the process parameters on the part radius,…”?
  6. Page 11, Table 5: Please indicate the units of Die radius /APR-P and Die radius /APRA-SFP.
  7. It is recommended that the descriptions of the unit be changed from 150 [mm] to 150 mm on page 4, 12 [%] to 12 % and 39 [%] to 39 % on page 6, and 16,25 [%] to 16,25 % on page 9 etc. throughout the article.
  8. The reviewer suggests that a schematic diagram is necessary to point out the Part Radius, Part height, and Die radius.
  9. Please explain how to determine the range of each parameter on Table 5.
  10. The conclusion should be shorter and more precise.

Author Response

Response to Reviewer 3 Comments

Point 1: It is recommended to point out which components are in Figure 3. In doing this, it will be easier for the reader to understand the meaning that the authors want to express.

Response 1: Figure 3 was updated as indicated; Components are now labelled on each image.

Point 2: Page 2: …improvements [7], [27].-> …improvements [7,27].

Response 2: The bibliographic references throughout all the text were revised and corrections have been made, as indicated.

Point 3: Page 7: …composite materials [39], [40].-> …composite materials [39,40].

Response 3: The bibliographic references throughout all the text were revised and corrections have been made, as indicated.

Point 4: Page 8, Table 3: Please indicate the difference between Die speed [Hz] and Die speed [mm/s]

Response 4: The “die speed [Hz]” was renamed as “die control frequency in Hz”, to avoid misunderstandings. We consider that if the die speed was indicated only in Hz it could be difficult to understand. The text that refers to Table 3 contains the reasons why die control frequency in Hz was necessary and indicates the necessity of correlating it to the speed in mm/s: “The correlation between die control frequency in Hz and die speed in mm/s is presented in Table 3 for a complete insight into the process”. Also, the Table 3 description was modified to avoid any confusion.

Point 5: Page 10: Please doule check Figures 10 and 11. Are both “Results of the ANOVA analysis highlighting the influence of the process parameters on the part radius,…”?

Response 5: The description of Figure 11 was changed, indicating the part height and not the part radius.

Point 6: Page 11, Table 5: Please indicate the units of Die radius /APR-P and Die radius /APRA-SFP.

Response 6: The APR-P and APR-ASFP are now explained on page 7, starting with line 185, in terms of radius measuring methodology, when using the GOM system. The text that refers to Table 5, at page 11, from line 288, was modified and additional indications were added: “A simple method was used to compare the parts; the die radius was divided by the average part radius (APR), obtaining a deformation coefficient”; this coefficient is dimensionless and is referred in Table 5 as “Die radius /APR-P coefficient” and “Die radius /APRA-SFP coefficient”.

Point 7: It is recommended that the descriptions of the unit be changed from 150 [mm] to 150 mm on page 4, 12 [%] to 12 % and 39 [%] to 39 % on page 6, and 16,25 [%] to 16,25 % on page 9 etc. throughout the article.

Response 7: The description of the units was changed through the text, removing the square brackets.

Point 8: The reviewer suggests that a schematic diagram is necessary to point out the Part Radius, Part height, and Die radius.

Response 8: Figure 7 was modified to include the overall dimensions of the die, also indicating the part height and part radius.

Point 9: Please explain how to determine the range of each parameter on Table 5.

Response 9: The description of Table 5 was updated to include information regarding part radius (“APR-P” and “APR-ASFP”) and height (“Part height, programmed” and “Part height, ASFP”), and represent the measurements made using the GOM system, deformation coefficient (“Die radius /APR-P coefficient” and “Die radius /APRA-SFP coefficient”) and process time value, that are explained starting with line 286.

Point 10: The conclusion should be shorter and more precise.

Response 10: Future directions of research were removed from this section. The conclusions now refer to the overall results of the experimental research.

Round 2

Reviewer 1 Report

I did spend some time looking at what has been published in this journal and stand by my initial recommendation of this belonging in another journal.  

I believe the editors would like to override this, which is fine.